# Isolation may select for earlier and higher peak viral load but shorter duration in SARS-CoV-2 evolution

Junya Sunagawa[1,16], Hyeongki Park[2,16], Kwang Su Kim[2,3,4,16], Ryo Komorizono[5], Sooyoun Choi[2,4], Lucia Ramirez Torres[2], Joohyeon Woo[2], Yong Dam Jeong [2,4], William S. Hart[6], Robin N. Thompson [6,7,8], Kazuyuki Aihara[9], Shingo Iwami [2,10,11,12,13,14,17] ✉ & Ryo Yamaguchi [1,15,17] ✉

During the COVID-19 pandemic, human behavior change as a result of non-pharmaceutical interventions such as isolation may have induced directional selection for viral evolution. By combining previously published empirical clinical data analysis and multi-level mathematical modeling, we find that the SARS-CoV-2 variants selected for as the virus evolved from the pre-Alpha to the Delta variant had earlier and higher peak in viral load dynamics but a shorter duration of infection. Selection for increased transmissibility shapes the viral load dynamics, and the isolation measure is likely to be a driver of these evolutionary transitions. In addition, we show that a decreased incubation period and an increased proportion of asymptomatic infection are also positively selected for as SARS-CoV-2 mutated to adapt to human behavior (i.e., Omicron variants). The quantitative information and predictions we present here can guide future responses in the potential arms race between pandemic interventions and viral evolution.

The human impact of population densities and activities on the global environment has increased so dramatically that the current geological era has been termed the Anthropocene[1]. Human-mediated selection constitutes one of the most significant and pervasive selective pressures on Earth, changing at a pace that requires rapid evolution of adaptive responses by all organisms[2]. COVID-19-related restrictions and the resultant changes in human activities created a phenomenon termed "anthropause," that is, a considerable global slowing of human activities and the effects of human activity on nature[3].

Human history has always been interwoven with viruses. Viruses debilitate many people and can have large-scale demographic effects on human populations according to immunity and prior disease exposure[4]. On the other hand, humans are an essential arena in which viruses evolve. Changes in human population size, immunity, and

[1]Department of Advanced Transdisciplinary Sciences, Hokkaido University, Sapporo, Hokkaido, Japan. [2]interdisciplinary Biology Laboratory (iBLab), Division of Natural Science, Graduate School of Science, Nagoya University, Nagoya, Japan. [3]Department of Scientific Computing, Pukyong National University, Busan, South Korea. [4]Department of Mathematics, Pusan National University, Busan, South Korea. [5]Laboratory of RNA Viruses, Department of Virus Research, Institute for Life and Medical Sciences (LiMe), Kyoto University, Kyoto, Japan. [6]Mathematical Institute, University of Oxford, Oxford, UK. [7]Mathematics Institute, University of Warwick, Coventry, UK. [8]Zeeman Institute for Systems Biology and Infectious Disease Epidemiology Research, University of Warwick, Coventry, UK. [9]International Research Center for Neurointelligence, The University of Tokyo Institutes for Advanced Study, The University of Tokyo, Tokyo, Japan. [10]Institute of Mathematics for Industry, Kyushu University, Fukuoka, Japan. [11]Institute for the Advanced Study of Human Biology (ASHBi), Kyoto University, Kyoto, Japan. [12]Interdisciplinary Theoretical and Mathematical Sciences Program (iTHEMS), RIKEN, Saitama, Japan. [13]NEXT-Ganken Program, Japanese Foundation for Cancer Research (JFCR), Tokyo, Japan. [14]Science Groove Inc, Fukuoka, Japan. [15]Department of Zoology & Biodiversity Research Centre, University of British Columbia, Vancouver, BC, Canada. [16]These authors contributed equally: Junya Sunagawa, Hyeongki Park, Kwang Su Kim. [17]These authors jointly supervised this work: Shingo Iwami, Ryo Yamaguchi. ✉e-mail: iwami.iblab@bio.nagoya-u.ac.jp; ryamaguchi@sci.hokudai.ac.jp

behavior based on public health policy can facilitate the rapid evolution of viruses[5].

We are facing the ongoing rapid emergence and adaptation of SARS-CoV-2. The virus was initially discovered in Wuhan, China, in late December 2019 (B lineage, strain Wuhan-Hu-1) and spread worldwide. Virus bearing the D614G substitution on the surface of the spike protein emerged in 2020 and became dominant among the circulating SARS-CoV2 variants[6]. The Alpha variant (B.1.1.7), first detected in the UK at the end of 2020, displayed a 43–90% higher reproduction number than pre-existing variants[7]. Interestingly, however, Alpha faded away with the rise of the more transmissible Delta variant (B.1.617.2)[8,9], which was first documented in India in October 2020. Then, the Omicron variant (originally B.1.1.529), which was first isolated in South Africa on October 2021, spread aggressively. A range of Omicron subvariants (BA.1, BA.2, BA.4, and BA.5) eclipsed the Delta variant, and the Omicron variant became the predominant variant worldwide after February 2022[10]. Currently, several new Omicron subvariants (BQ.1, XBB, and others) are emerging and are reported to be more transmissible and resistant to immunity generated by previous variants or vaccinations[11]. Collectively, these observations indicate that SARS-CoV-2 has continuously been evolving and variants have continued to emerge to replace existing viral strains worldwide, as observed with influenza[12].

An understanding of the epidemiological and clinical characteristics of current and future emerging infectious diseases is important for developing adaptive treatments, including antivirals and vaccinations, and screening and isolation strategies. Thus, evaluating and predicting how viral dynamics changes throughout infection through evolution is essential[5,13–16]. An unprecedentedly large volume of high-quality data have accumulated during the COVID-19 pandemic, and here we analyze the data on SARS-CoV-2 variants to benefit preparedness for future pandemics. As we discuss elsewhere[15–18], quantifying and comparing the timing and height of peak viral load and the duration of viral shedding among SARS-CoV-2 variants are of critical importance. In addition, understanding the driving forces behind viral evolution is also required, given the different selective pressures acting on the virus[5]. In general, the infection- and/or vaccine-induced immune response to antiviral drugs (i.e., pharmaceutical interventions: PIs) leads to virus evolution. The strongest evidence for selection based on human intervention is the rapid evolution of immune-escaping mutations in the Omicron variant, which often occur in parallel and can be predicted ahead of time[19]. In contrast, non-pharmaceutical interventions (NPIs), such as isolation, quarantine, social distancing, and wearing a face covering, efficiently prevent close contact, in particular, in symptomatic patients, given that most cases are the result of community transmission. NPIs have proven effective in reducing the spread of SARS-CoV-2 in many contexts[20–25]. The introduction of strong selection pressure by NPIs and the evolutionary impact of this has been the focus of much attention[26–28], but how viral load dynamics are altered in vivo has not yet been explored. We were thus interested in the role of isolation as a strong NPI and explored the possible impact of such human behavioral changes on SARS-CoV-2 evolution.

So far, statistical analyses have shown that SARS-CoV-2 viral kinetics are mainly dominated by individual-level variation[29], but the kinetics may be partly determined by immunity and variant[30]. Here, to explore the isolation-driven viral evolution, we first quantified the viral dynamics from existing data on the viral load over the course of SARS-CoV-2 infection for the pre-Alpha (non-variants of interest/variants of concern [VOI/VOCs] types)[14,31], Alpha, and Delta variants (i.e., individual-level virus infection model) considering both individual and variant-specific variations in kinetics. Then, to explain and predict the evolutionary patterns of SARS-CoV-2 variants in terms of the time-series pattern of viral load, we developed a multi-level population dynamics model by coupling a population-level virus transmission model with the individual-level virus infection model. Under different clinical phenotypes for COVID-19 patients, defined here as the incubation period and the proportion of symptomatic infections, we evaluated how the time-series patterns of viral load and, therefore, SARS-CoV-2, evolve under isolation of infected individuals. We interestingly demonstrate that NPIs may select for earlier and higher peak viral load but shorter duration of infection in SARS-CoV-2. Although there are multiple ways to cause the recent evolutionary trajectory, the concepts and variables that affect the current transition can be anticipated. We discuss the potential evolution of SARS-CoV-2 as it adapts to maximize transmissibility in the presence of a human behavior change.

## Results

### Characterizing the time-series pattern of viral load for SARS-CoV-2 variants

We analyzed the viral load of SARS-CoV-2 for the pre-Alpha (non-VOI/VOCs)[14,31], Alpha[31,32], and Delta variants[31,32] using a simple mathematical model describing viral dynamics[15–18] (see Table S1 and METHODS). To consider inter-individual variation in the patients' viral loads, we employed nonlinear mixed effect models to estimate the parameters. The estimated population parameters are shown in Table S2. The typical behavior of the mathematical model, Eqs. (4, 5), using these parameters is shown in Fig. 1a and Fig. S1 for the pre-Alpha (black), Alpha (blue), and Delta variants (red), respectively. Note that the time of infection (i.e., $t=0$) is estimated in our model fitting by using fixed initial values[33] (see METHODS). These quantified virus infection dynamics highlight how the properties of the virus differ variant by variant.

To characterize and compare the infection dynamics of these SARS-CoV-2 variants, we quantified the distributions of the peak viral load, peak time, and duration of viral shedding, $D$, (Fig. 1b–d). Here duration of viral shedding indicates the time for which the viral load is detectable. We found that peak time did not differ significantly between the pre-Alpha and Alpha variants (Fig. 1c). The peak viral load also was not significantly different between the Alpha and Delta variants (Fig. 1b, d). However, interestingly, we found that the peak viral loads for the Alpha and Delta variants were higher than for the pre-Alpha variants ($p = 2.9 \times 10^{-7}$ and $8.8 \times 10^{-6}$ by the Mann-Whitney-Wilcoxon test, respectively) (Table 1). The peak time and duration of viral shedding for the Delta variant were shorter than those for the pre-Alpha and Alpha variants (for the peak time: p-values are less than $2.2 \times 10^{-16}$ for the duration of viral shedding: $p = 1.4 \times 10^{-15}$ and $1.1 \times 10^{-6}$ by the Mann-Whitney-Wilcoxon test, respectively) (Table 1). Taken together, our results highlight that the duration of viral shedding was shortened and the peak viral load was increased and advanced as SARS-CoV-2 evolved. This implies that SARS-CoV-2 is evolving to a more "acute phenotype"[34], mainly characterized by an advanced peak time and a shorter duration of viral shedding compared with the pre-Alpha variants.

To further characterize the time-series pattern of viral load for SARS-CoV-2 variants during evolution of the virus (i.e., the Alpha variant emerged during the spread of the pre-Alpha variants, and the Delta variant emerged after the spread of the Alpha variant), we first investigated the relation between the duration of viral shedding, $D$, and the cumulative log-transformed viral load, $V_{total}$, defined as the area under the curve (i.e., AUC [$\log_{10}$ viral RNA copies/ml × days]) (Fig. 1e). We found $V_{total}$ increased as $D$ increased (i.e., positive correlation) (see METHODS). Interestingly, there was also a trend that $V_{total}$ decreased as SARS-CoV-2 evolved: $V_{total}$ for the Delta variant was smaller than that for the pre-Alpha and Alpha variants ($p = 1.1 \times 10^{-7}$ and $9.4 \times 10^{-6}$ by the Mann-Whitney-Wilcoxon test, respectively). In addition, to understand the epidemiological consequences of these variants, we also calculated the transmission probability of each variant depending on the time after infection (see Fig. 1f and METHODS).

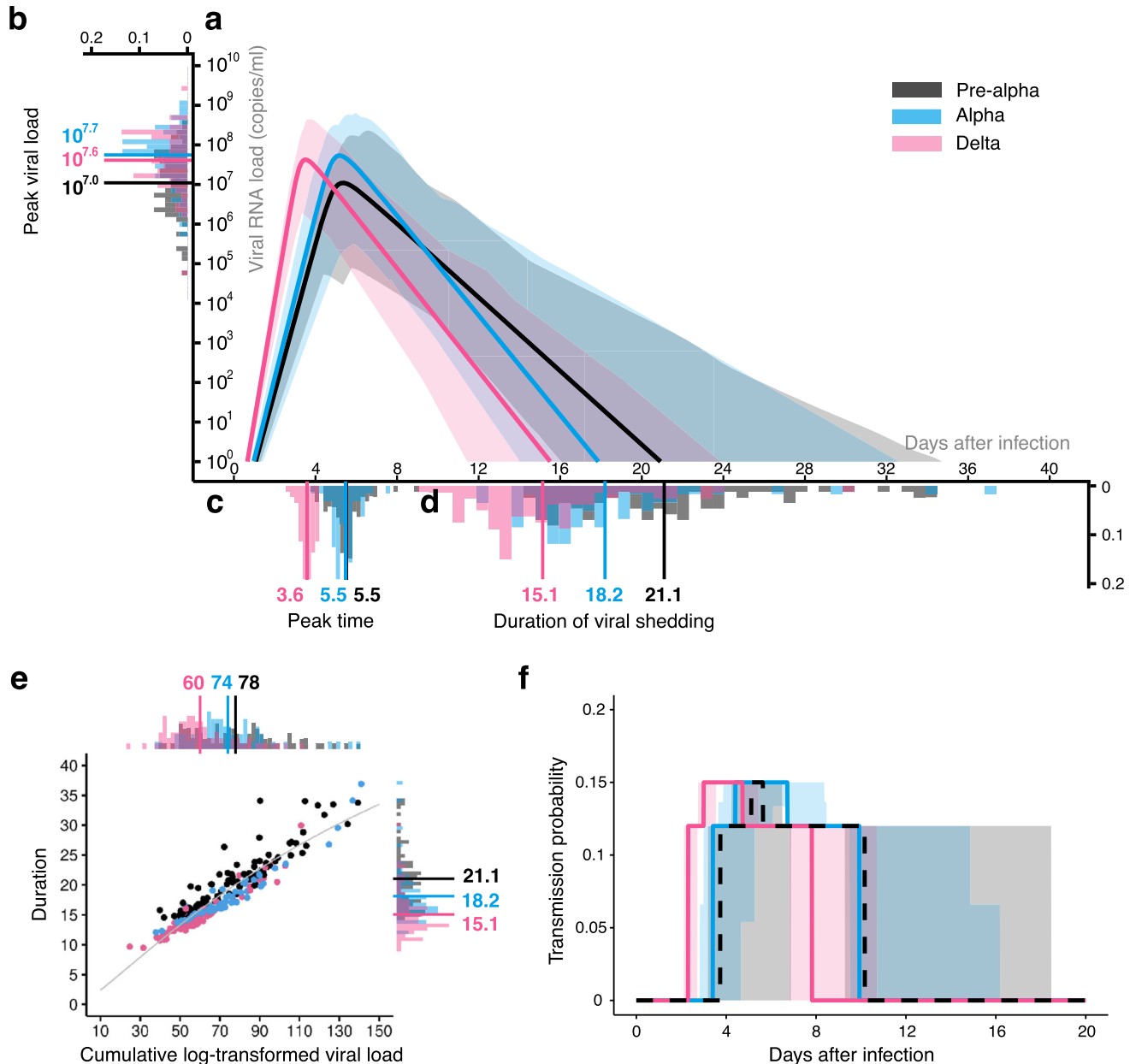

**Fig. 1 | Quantification of evolving SARS-CoV-2 infection dynamics.** Viral load data from upper respiratory specimens (i.e., nose and pharynx) of 86 patients infected with the pre-Alpha (non-VOI/VOCs) variant, 59 Alpha variant patients, and 80 Delta variant patients were used for parameter estimation of the viral dynamics model. **a** The inferred viral dynamics for pre-Alpha (black), Alpha (blue), and Delta (red) variants are plotted. The solid curves correspond to the solution of Eqs. (4, 5) using the best-fit population parameters, and the error bands represented as shadow regions indicate the 95% interquartile range of the inferred viral load interval for all individuals at each time point. The distribution of the peak viral load, peak time, and duration of viral shedding for each patient are also shown in **b**, **c**, and **d**, respectively. Solid bars are their mean values. **e** The relation between the duration of viral shedding and the cumulative log-transformed viral load is shown. The gray curve is the best-fitting curve for Eq. (7). The distributions illustrated alongside the top and right parts are for the cumulative viral load and the duration of viral shedding, respectively, and the bars are their mean values. **f** The time-dependent transmission probability of each variant is calculated. In all panels, pre-Alpha, Alpha, and Delta variants are colored in black, blue, and red, respectively.

**Table 1 | Summary of viral load properties for SARS-CoV-2 variants**

| Variant | Peak viral load | Peak time | Duration of viral shading | Cumulative log-transformed viral load ($V_{total}$) |
|---|---|---|---|---|
| Pre-Alpha | $10^{7.0a}(10^{4.8} - 10^{8.6})^b$ | 5.5(3.9 − 9.1) | 21.1(14.6 − 34.1) | 78.3(39.8 − 139.3) |
| Alpha | $10^{7.7}(10^{5.8} - 10^{9.1})^c$ | 5.5(3.9 − 10.7) | 18.2(12.1 − 36.9)$^c$ | 74.4(37.8 − 141.1) |
| Delta | $10^{7.6}(10^{4.8} - 10^{9.4})^c$ | 3.6(2.7 − 4.2)$^c$ | 15.1(9.5 − 30.0)$^c$ | 60.4(24.7 − 110.8)$^c$ |

$^a$Mean value.
$^b$Min-Max value.
$^c$Statistically different from pre-Alpha variant ($p = 0.01$ by the Mann-Whitney-Wilcoxon test

We found that the transmission probability of the Alpha variant remained a high transmission probability for a longer period of time than the pre-Alpha variant. Furthermore, the transmission probability of the Delta variant peaked at 3.0 days after infection[9] but decreased to almost 0 by 8.0 days, suggesting tFhe evolutionary process promotes effective and rapid transmission within human populations (see below for details).

### Prediction on time-series pattern of viral load of SARS-CoV-2 evolution with NPIs

In the comparison of viral shedding dynamics, the Alpha variant showed a higher peak viral load than the pre-Alpha variants (Fig. 1b). Although the duration of viral shedding for the Alpha variants was less than that for the pre-Alpha variant (Fig. 1d), the transmission probability of the Alpha variants remained higher for a long time (Fig. 1f). Note that a similar difference between the pre-Alpha and Delta variants can be observed (Fig. 1b, d, f). These findings demonstrate that the Alpha-like and the Delta-like variants showed a high peak viral load but a relatively short duration of viral shedding as the virus evolved among human populations.

In contrast, in a comparison of the Alpha and Delta variants, the selective force to facilitate the evolution of the Delta-like variant is not trivial. This is because the peak viral load and the duration of viral shedding were similar between the Alpha and Delta variants (Fig. 1b, d), and therefore the highest transmission probability maintains almost the same level and length for these variants (Fig. 1f). The difference between the Alpha and Delta variants is the peak time, that is, the peak time of the Delta variant was significantly earlier than for the Alpha variant (Fig. 1c). Our hypothesis is that the Delta-like variants showing an earlier peak time but similar peak viral load and duration of viral shedding (i.e., acute phenotype) had a general advantage under human-mediated selection pressures (i.e., earlier infections result in more descendants). COVID-19 patients show both symptomatic and asymptomatic infection, and the incubation period and the proportion of asymptomatic patients are decreasing and increasing, respectively, as SARS-CoV-2 evolves[35,36]. To prevent the chain of transmission, patients showing symptoms are usually isolated or hospitalized or subjected to other NPIs[37,38]. Symptomatic patients primarily generate secondary infections prior to symptom onset, that is, during the pre-symptomatic period. On the other hand, asymptomatic patients can be infectious during the whole duration of viral shedding. Below we show that a strong NPI, defined as the isolation of symptomatic infected individuals after symptom onset, could be a driving force for SARS-CoV-2 evolution. Note that although we can easily incorporate variant-specific virological characteristics such as infectivity per virus by changing the probability of transmission (see METHODS), we ignore these differences here for simplicity.

### NPI-driven SARS-CoV-2 evolution

Using the probabilistic multi-level model, which considers both individual-level virus infection dynamics and population-scale transmission, as shown in Eqs. 1–3, 6, and the genetic algorithm (GA), which mimics virus phenotypic evolution[34], we explored how the time-series patterns of viral load evolved under NPIs. The schematic diagram of our multi-level modeling is described in Fig. 2, and a detailed explanation is provided in the METHODS.

Briefly, from the assumption that the virus evolves to increase its "transmission potential", defined as $R_{TP}$ (i.e., the total number of secondary cases generated throughout the infectious period), we calculate $R_{TP}$ under various patterns of infection rate, $\beta$, and virus production rate, $p$. Thus, the transmissibility fitness landscape of $R_{TP}$ is constructed as a function of ($\beta,p$), given the incubation period (i.e., time from infection to symptom onset), $T^*$, and the proportion of symptomatic patients, $f$. Note that the transmissibility fitness landscape of $R_{TP}$ is constructed as a function of only $\beta$ and $p$, because the

other parameters or initial values are estimated or determined (see METHODS). Under this analysis, we considered the (additional) effect of isolation, which is only applicable to symptomatic individuals and which assumes that isolation always perfectly prevents the transmission chain (i.e., perfect isolation; $T^* < t$). In other words, the (baseline) effect of isolation, which is applicable to all individuals regardless of symptomatic or asymptomatic infection (e.g., wearing a face mask, social distancing), is assumed to be involved in the parameters in our multi-level model without loss of generality.

In the absence of symptomatic infection (i.e., all patients are asymptomatic: $f = 0$), we analyze SARS-CoV-2 evolutionary dynamics by exploring the optimal set of $\beta$ and $p$, which characterizes the time-series pattern of viral load, with GA (see Table S3, S4 and Algorithm S1). Without symptomatic infection, there is no effect of symptom-dependent isolation on the transmission chain at all. The optimal sets of ($\beta,p$), the transmissibility fitness landscapes, the trajectories of $R_{TP}$ along the course of GA, and its distribution are described in Fig. S2a–c, respectively. The time-series patterns of viral load with the optimal parameters and the corresponding timing of peak viral load (i.e., peak time) are described in Fig. S2d, e, respectively.

In contrast, in the absence of asymptomatic infection (i.e., all patients are symptomatic; the fraction of symptomatic infection is $f = 1$), all individuals lose their transmissibility by NPIs after symptom onset ($T^* < t$). Depending on the incubation period, $T^*$, the distributions of optimal sets of ($\beta,p$) are located in different parameter ranges, and the corresponding transmissibility fitness landscapes are altered (Fig. S3a, b). Because isolation can prevent the transmission chain before the focal infected individuals became highly infectious, given a small $T^*$ such as $T^* = 1.0$ day (i.e., almost no transmissions occur during the pre-symptomatic period), $R_{TP}$ with the optimal parameters is basically less than 1 (Fig. S3b). On the other hand, given a large $T^*$ (e.g., $T^* = 10.0$ days), isolation is delayed, and therefore the transmissions are allowed during the pre-symptomatic period. Thus, $R_{TP}$ increases as $T^*$ increases (Fig. S3b). Note that the effect of isolation is weak for $T^* = 10$ because the viral load already drops to a very low level at the moment when isolation is applied to the symptomatic patients, and thus the transmission probability is low. In fact, the virus evolution shows similar patterns observed in the absence of symptomatic infection ($f = 0$) (Fig. S2a–c vs. Fig. S3a–c for $T^* = 10.0$, for example). The time-series patterns of viral load and the corresponding timing of peak viral load are also described in Fig. S3d, e, respectively. Interestingly, compared with the case where all patients are asymptomatic ($f = 0$), acute phenotypes showing an earlier peak but similar peak viral load and duration of viral shedding are selected (i.e., isolation-driven evolution). Thus, in the absence of asymptomatic infection, selection generated by isolation is maximized and drives convergent evolution of viral load dynamics (Fig. S3d). Additionally, because of smaller transmission potential (i.e., secondary [focal] infectious individuals may not be effectively generated and sometime less than one), isolation-driven evolution might be unlikely to occur under this situation.

In the realistic situation, there are both asymptomatic and symptomatic infections (i.e., $0 < f < 1$), and only symptomatic patients interfere with secondary infection by isolation after symptom onset. We next evaluated virus evolution under the different incubation period of $T^*$ with the fixed fraction of $f = 0.3$ (i.e., 30% of infected individuals show symptoms). Note that there is substantial uncertainty in the value of $f$, and we conduct supplementary analyses for different values of $f$ (see below). As described in Fig. 3a, the optimal sets of ($\beta,p$) are widely distributed for $T^* = 1.0$ and 3.0 days, but those are narrowed for $T^* = 6.0$ and 10.0 days. This is because the transmissibility fitness landscapes of viruses differ between asymptomatic and symptomatic patients in the case of relatively small incubation periods, whereas the difference is small for large periods (Fig. 3b). We note that having a higher peak viral load while maintaining a rather long duration favors

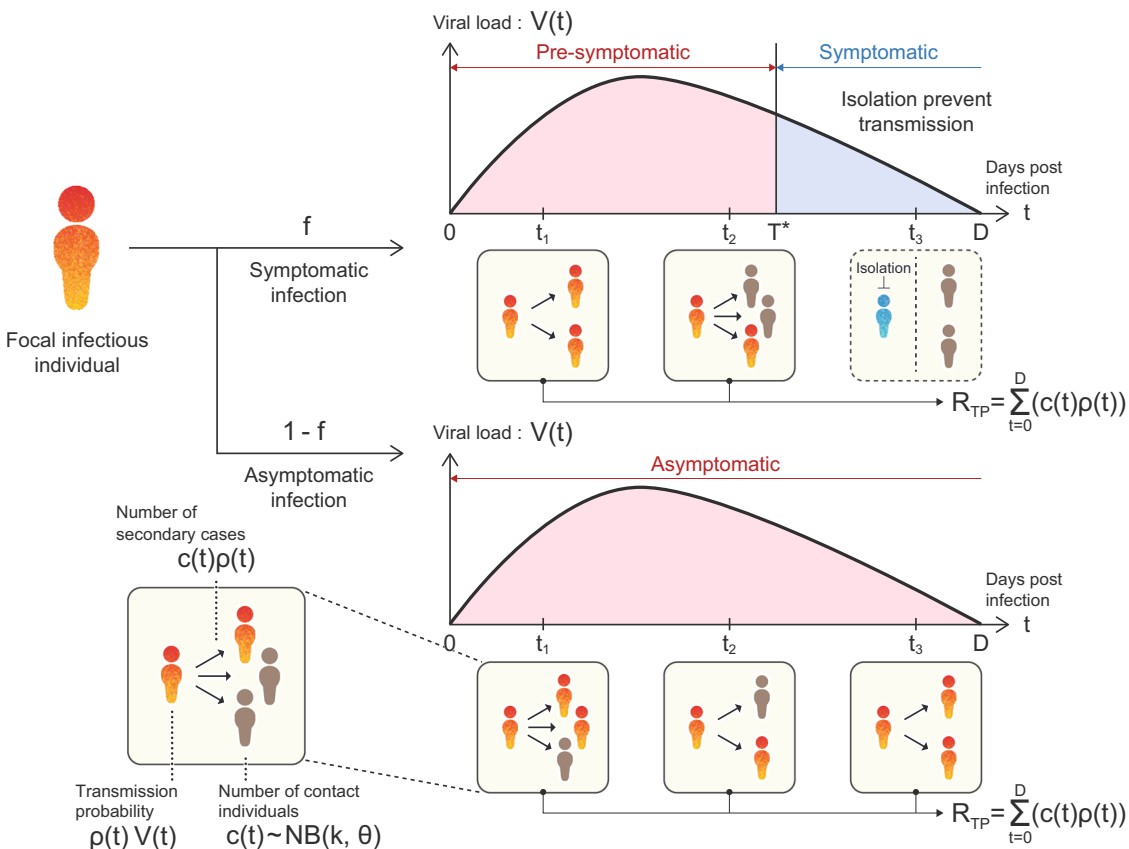

**Fig. 2 | Schematic of multi-level disease transmission.** A schematic of the multi-level population dynamics model is depicted. Virus transmission occurs from an infected individual to susceptible individuals depending on the infected individual's infectivity, which depends on their viral load. At each time step (i.e., day), the focal infected individual has contact with multiple susceptible individuals. Here the contact numbers are assumed to follow a negative binomial distribution, which does not depend on the viral load. The shape parameters $k$ and $\theta$ for this binomial distribution are presented in Table S3. The sum of newly infected individuals (i.e., [$c(t)$: number of contacted individuals per day (everybody is susceptible)] × [$\rho(t)$:

transmission probability per contacted individual]) during the infectious period is calculated as $R_{TP}$, called the "transmission potential". Note that the focal infectious individuals are assigned one of two properties: symptomatic or asymptomatic. Whether being symptomatic or asymptomatic is randomly assigned by a constant probability $f$ or $1-f$, respectively. It is assumed that the transmission chains from the focal symptomatic infectious individuals are perfectly inhibited by isolation after symptom onset, $T^* < t$. Otherwise, $R_{TP}$ is calculated using the whole duration of viral shedding of the focal asymptomatic infectious individual. Thus, $R_{TP}$ for the symptomatic and asymptomatic individuals are different here.

an increase in $R_{TP}$, and viral dynamics with this characteristic are achieved when $\beta$ is low and $p$ is relatively high, such as optimal sets of $(\beta, p)$ for the asymptomatic case (or in the absence of isolation) in Fig. 3b.

In the search for the optimal parameter set by GA, depending on the type of focal infectious individual (who is stochastically chosen), directions of virus evolution stochastically change so that neither peak of the transmissibility fitness landscape can be unreachable (compare the colored dots in Fig. 3a with the white dots in Fig. 3b for $T^* = 1.0$ and 3.0) unless the evolutionary rate is extremely fast. Thus, an intermediate parameter set becomes the equilibrium state. Moreover, we calculated $R_{TP}$ for each infected individual: the trajectories of $R_{TP}$ with the optimal parameters vary around 1, especially with short incubation periods (Fig. 3c), because $R_{TP}$ depends on the types of focal infectious individual and $R_{TP}$ for the symptomatic individuals is basically smaller than that for the asymptomatic individuals (sometimes less than 1) because of isolation (Fig. 3b).

As observed in the absence of asymptomatic infection, the time-series patterns of viral load (Fig. 3d) with the optimal parameters (Fig. 3e) also depend largely on the incubation period $T^*$ because of isolation. Of note, in the realistic situation with $0 < f < 1$, because (secondary) infectious individuals are mainly maintained via transmissions by asymptomatic individuals, isolation-driven evolution for relatively small incubation periods may occur, and the peak viral load is

advanced. Fig. 3d, e also demonstrate that faster peak time is associated with larger value of the product of $p$ and $\beta$. This is due to the combined effect of larger $p$ and $\beta$ values, resulting in both rapid and increased amount of virus production during an early phase of infection. For our sensitivity analysis, we conducted the same simulations for $f = 0.5, 0.6, 0.7$ and obtained similar conclusions (see Fig. S4).

We have thus far assumed 'perfect isolation' in our results, where transmission is completely halted once patients become symptomatic. However, we understand that in practice, isolation is sometimes imperfect, leading to continued virus transmission. To account for this, we introduced a factor $\xi$, modifying the contact numbers post symptom onset ($C^*(t) = C(t) \times \xi$), with $\xi$ representing incomplete isolation probabilities of 0.1 and 0.3. In general, this extension yielded consistent outcomes with our initial findings (Fig. S5). However, it indicated that incomplete isolation leads to a less pronounced evolution towards an earlier, higher peak in viral load dynamics at a small $T^*$ (such as $T^* = 1$: it is unreasonably small in the context of COVID-19 and therefore we may exclude this situation thought) (Figs. S5–1). Even in cases where $T^*$ is small, and it significantly drives virus evolution, the deterministic selective pressure is weakened due to the probabilistic failure of isolation. This situation is analogous to a scenario where all patients have asymptomatic infections as a result, making advancing the peak viral load no longer beneficial because of the short pre-symptomatic period. The trend becomes more prominent when the

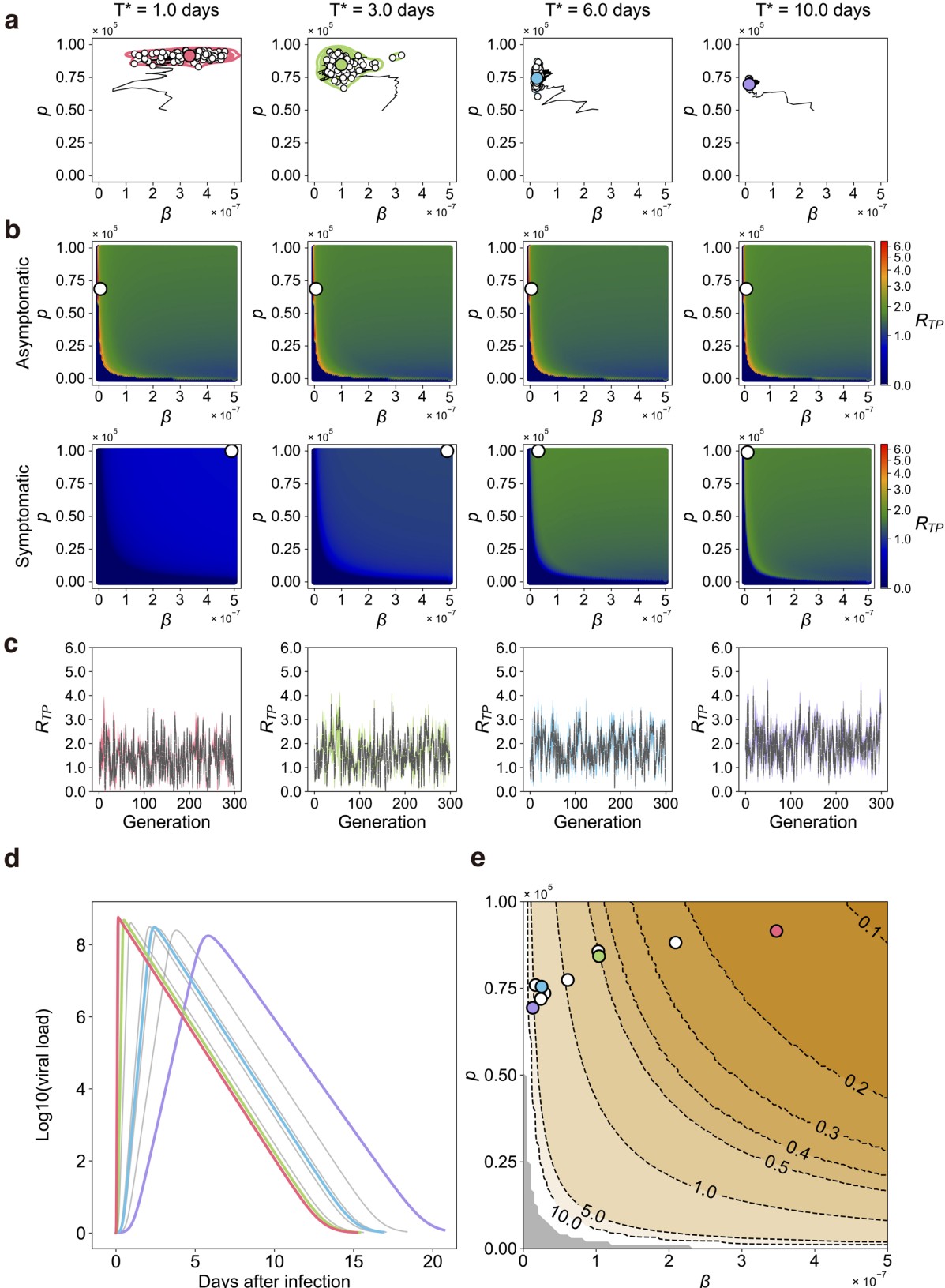

probabilistic failure of isolation occurs frequently (e.g., $\xi = 0.3$ in Figs. S5–2).

As we confirmed in Figs. 3, Fig. S2, S3, and S4, the higher proportion of symptomatic infection enhances the selection pressure for virus evolution via isolation. Taken together, these results suggest that while isolation inhibits the chain of transmission from symptomatic

focal infectious individuals after symptom onset, in the realistic situation, SARS-CoV-2 evolves toward the acute phenotypes.

**Effect of prior immunity on NPI-driven SARS-CoV-2 evolution**

Considering another modeling assumption, our initial mathematical model did not account for the influence of COVID-19 vaccinations or

**Fig. 3 | SARS-CoV-2 evolution in silico ($f$ = 0.3). a** Genetic algorithm (GA) exploring the evolutionary trajectories on the ($\beta,p$) plane until the generation of 300 is applied, depending on different values of the incubation period, $T^*$. Here $\beta$ and $p$ are the infection rate and the virus production rate, respectively. All symptomatic individuals lose their transmissibility by isolation after symptom onset ($T^* < t$). The white dots represent the endpoint of 100 independent simulation runs, and the contour lines are the kernel density estimation of their distribution. The colored dot in each panel is the mean value of the white dots, which represents the set of evolutionary outcomes of ($\beta,p$) under the parameters we used. The colors used here are independent to them in Fig. 1. The black line is the mean trajectory of the GA through 300 generations. **b** The mean transmissibility fitness landscapes aggregated solely from the asymptomatic (top row) and symptomatic (bottom row) individuals are described, respectively, using 100 runs of GA. The white dot represents the maximum value of the mean transmissibility fitness, $R_{TP}$. Note the asymptomatic individuals have larger transmissibility fitness than symptomatic cases on average, regardless of $T^*$ values. **c** The trajectories of $R_{TP}$ along the course of GA with different $T^*$ are calculated. The gray dotted lines are the mean trajectory over 100 trials of colored lines. **d** The time-series patterns of viral load with the optimal parameters of ($\beta,p$) with different $T^*$, which were obtained in **a**, are shown. **e** The contour-plot for the timing of peak viral load (i.e., peak time) is shown. Each curve is colored accordingly. The gray region is the parameter range satisfying $R_{TP} < 1$.

prior infections. Given the widespread infection and vaccination leading to high-level population immunity worldwide, this factor could have a profound impact on the evolution of the virus. Some studies suggest that COVID-19 vaccinations may affect the SARS-CoV-2 infection dynamics[32,39]. Accordingly, we conducted further analysis to investigate the impact of prior immunity caused by vaccinations or prior infections on our findings.

We first reanalyzed SARS-CoV-2 infection dynamics in Fig. S6, including additional 68 participants (14 and 54 patients for the Alpha and the Delta variants, respectively), who were excluded due to prior infections, and obtained the consistent statistical trends in line with our findings described in Fig. 1a, reinforcing the robustness of our conclusions. Specifically, we observed that the duration of viral shedding keeps decreased in evolution. The peak time did not exhibit significant differences between the pre-Alpha and Alpha variants, and the peak viral load did not significantly differ between the Alpha and Delta variants. However, the peak viral loads for both the Alpha and Delta variants were higher compared to the pre-Alpha variants ($p = 0.002$ and $3.3 \times 10^{-7}$, respectively). Moreover, Delta variant exhibited a shorter peak time in comparison to both the pre-Alpha and Alpha variants ($p = 2.2 \times 10^{-16}$ for both cases) (Fig. S6). These confirmed that the statistical trend for SARS-CoV-2 evolution is remained regardless of the prior infections.

Furthermore, we extended our in silico analysis to incorporate a vaccinated population, considering vaccination rates set probabilistically at 50% and 80%. For those vaccinated, we increased the clearance rate ($\delta$) by 1.5 times. This assumption for clearance rate shortened the viral shedding period by about 3 days, and this is larger than the previously reported shortening period of 2 days[32]. Thus, a 1.5-fold increase in clearance rate is sufficient to account for an increase in clearance rate caused by the vaccination. The incorporation of this stochastic environment led to a modest increase in the variance of optimal virus parameters. However, it is essential to note that these changes in the model maintained our main conclusion. The mean values derived remained consistent with our original findings: viruses were selected for had an earlier peak and higher viral load dynamics, but a shorter duration of infection (Fig. S7). The selective pressure for increased transmissibility shapes the viral load dynamics, and isolation measures are likely to be a critical driver of these evolutionary transitions even considering the prior immunity.

### Validating SARS-CoV-2 evolution with Omicron variants

The Omicron variant (originally B.1.1.529) was first isolated in South Africa in October 2021[10,40]. After that time, of the identified Omicron subvariants, the BA.1 subvariant rapidly spread worldwide and became the most prevalent SARS-CoV-2 variant in many countries (Fig. 4a). Here, we consider simply the situation that BA.1 emerged during the spread of the Delta variant, which was the most predominant variant worldwide before BA.1, under NPIs.

To validate our hypothesis that isolation may be a selection pressure to further drive the evolution of viral phenotypes, we additionally used the longitudinal viral load of SARS-CoV-2 from the BA.1 subvariant in 49 infected patients[41] (Table S1), and similarly analyzed those data together with the same data for the Delta variant (see METHODS in detail). The individual viral loads for the Delta variant (red) and BA.1 subvariant (green) are described in Fig. S8, and the estimated population parameters are shown in Table S5. We found that the time-series pattern of viral load for the BA.1 subvariant differed from that for the Delta variant as described in Fig. 4b. We further characterized and compared the distributions of the peak viral load, peak time, and duration of viral shedding in Fig. 4c. Interestingly, while we found almost the same peak time for the BA.1 subvariant viral load with a similar duration of viral shedding ($p = 0.09$ and 0.48 by the Mann-Whitney-Wilcoxon test, respectively), the comparison of peak viral load suggested a lower peak for the BA.1 subvariant than for the Delta variant ($p = 1.3 \times 10^{-6}$ by the Mann-Whitney-Wilcoxon test). On the other hand, as we expected, the BA.1 subvariant showed an earlier peak time and shorter viral shedding compared with the pre-Alpha and Alpha variants (see Fig. 1a–d vs. Fig. 4b, c). Altogether, these results imply that SARS-CoV-2 evolution toward an earlier peak viral load was maintained but slowed down. Interestingly, similar evolutionary trends were confirmed in a non-human primate model infected with different VOCs of SARS-CoV-2[42], obviously without NPIs in these cases. We discuss a mechanism behind those convergent phenotypic changes in detail later (see Discussion). Our quantitative empirical clinical data analysis based on our hypothesis could support that SARS-CoV-2 variants have evolved their acute phenotypes via a human-mediated selection pressure such as isolation.

## Discussion

Understanding the entire arc of a virus's impacts on an outbreak requires knowledge of how the virus and human populations interact and placing the relationship within not only immunological but societal and cultural contexts. Here we quantified the infection dynamics of the SARS-CoV-2 variants from the patients' longitudinal viral load data. In the process of evolutionary transitions among the pre-Alpha, Alpha, Delta, and Omicron variants, we found the durations of viral shedding were shortened, except for the Omicron variant, and the peak viral load was increased (from the pre-Alpha to Alpha variants) and advanced (from the Alpha to Delta variants) (Fig. 1a–d). Increasing peak viral load (unless individuals become so sick that they do not circulate) is obviously advantageous for virus evolution because it effectively increases transmission probability (Fig. 1f). In contrast, we interestingly demonstrated that the advantage of the advancing peak viral load may depend on clinical phenotypes (i.e., the incubation periods: $T^*$ and the proportion of symptomatic: $f$) for virus variants and the community environment (i.e., with/without isolation) (Fig. 3, Figs. S2 and S3). Hence, using quantitative empirical data analysis and the multi-level population dynamics model, we suggested that isolation potentially plays the role of a human-mediated selection pressure for SARS-CoV-2 evolution.

A higher proportion of asymptomatic infection and a longer incubation period than for other acute respiratory viruses are notable features of COVID-19 and are major factors increasing the basic/effective reproduction number[43,44]. The proportion is reported to be around 30%[45,46], and the incubation period is 2 to 12 days[47]. Because

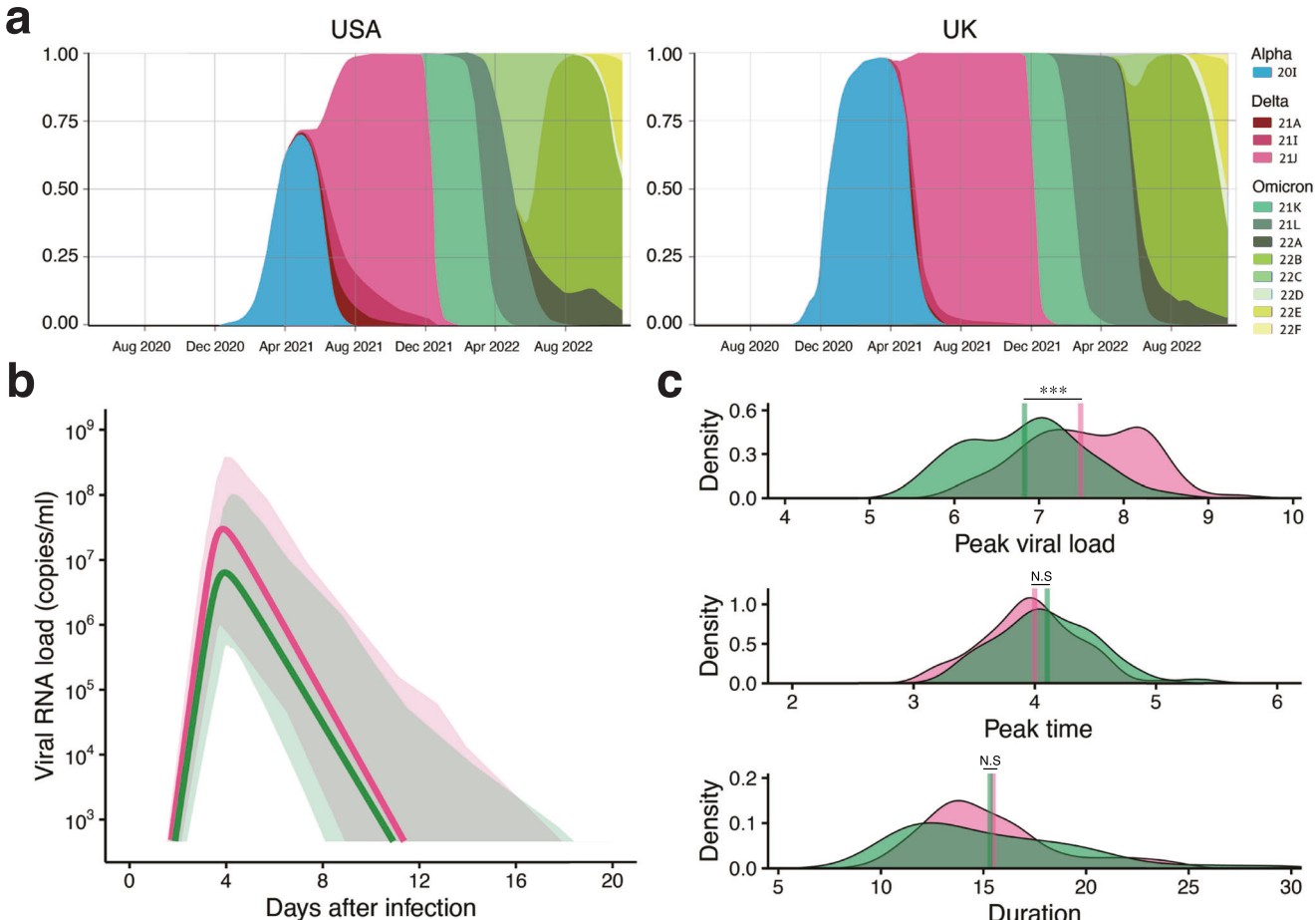

**Fig. 4 | Quantification of SARS-CoV-2 BA.1 subvariant infection dynamics. a** An overview of frequency changes in SARS-CoV-2 variants based on data from CoV-ariants.org is shown. The left and right panels show the trend in the United States and the United Kingdom, respectively, as examples (because the data we used here were from the USA and UK). See the labels for each color curve corresponding to each VOC. **b** The estimated viral dynamics using our model and publicly available data for the BA.1 subvariant (green) are described along with the Delta variant (red) using the best-fit population parameters. The solid curves correspond to the solution of Eqs. (4, 5) using the best-fit population parameters, and the error bands represented as shadow regions indicate the 95% interquantile range of the inferred viral load interval for all individuals at each time point. **c** The distributions of the peak viral load, peak time, and duration of viral shedding for each patient are shown with their mean values (vertical solid bars). A statistically significant difference between two groups was found only for peak viral load (p-values by the two-sided Mann-Whitney-Wilcoxon test are $1.3 \times 10^{-6}$ for peak viral load, 0.09 for peak time, and 0.48 for duration).

transmission via symptomatic individuals is usually prevented by isolation (and other PIs/NPIs), SARS-CoV-2 is mainly transmitted during the asymptomatic and pre-symptomatic periods, implying that advancing the peak viral load can effectively generate more secondary infections. As we explained in Fig. 3, under isolation, early peak viral loads such as for the Delta and Omicron variants are beneficial to variants having a shorter incubation period, because the effect of isolation on reducing transmission probability is mitigated and therefore $R_{TP}$ increases (see Fig. S9). Thus, in addition to PIs, isolation could be one of the driving forces promoting virus evolution as the Delta variant dominated after the spread of Alpha. In fact, broken down by VOCs, the incubation period of COVID-19 was 6.65 days for the pre-Alpha, 5.00 days for the Alpha, 4.50 days for the Beta, 4.41 days for the Delta, and 3.42 days for the Omicron variants[36,48]. The incubation period decreased as SARS-CoV-2 mutated, implying the selection pressures by NPIs enhanced virus evolution, as we confirmed in Fig. 3.

In contrast, when comparing the recently emerged Omicron variant with other variants, it has been reported that the Omicron variant is evolving to show a higher proportion of asymptomatic infection[35]. Whereas a shorter incubation period enhances an earlier peak viral load as discussed above, a higher proportion of asymptomatic infection impairs the driving force of virus evolution, meaning a simple one-

directional virus evolution (i.e., earlier and higher peak viral load) does not occur. Interestingly, these trends are observed for the subvariant BA.1 in Fig. 4b, c, that is, decreasing the peak viral load. From the perspective of virus evolution, which maximizes the basic/effective reproduction number (or an average of the transmission potential for symptomatic and asymptomatic infection $R_{TP}$ in this study, for example), the evolution of a higher proportion of asymptomatic infection is always favored because transmissions are not limited when those infected individuals are not isolated. This means that once the higher proportion of asymptomatic infection evolves, other viral phenotypic changes designed to avoid isolation will be suppressed. In other words, viral phenotypes showing slower and/or lower peak viral load can evolve (see Fig. S9). This perspective grounded on the assumption that the evolution of virus phenotype and the evolution of clinical phenotype may often be decoupled. While we believe that this assumption can be accepted[49,50], we also acknowledge that it should be noted that a consensus applicable to all cases has not been established[49,51,52].

A reasonable expectation during the COVID-19 pandemic is that the virus could evolve to develop increased transmissibility, reflecting adaptations to propagation in the new human host[5] and the rapidly changing environments by PIs and NPIs. Taken together, as described in Fig. S9, we found there are at least two antagonistic forces

enhancing and mitigating the earlier and higher peak viral load evolution via isolation during the process of SARS-CoV-2 evolution from the pre-Alpha to the Omicron variants via the Alpha and Delta variants: decreasing the incubation period and increasing the proportion of asymptomatic infection, respectively. Although both forces may increase transmissibility, the balance between these two forces mitigates or stops the evolution of SARS-CoV-2 phenotypes defined as the time-series patterns of viral load. There is a possibility that the strength of NPIs per se could be weakened worldwide as interest in COVID-19 drops off, and therefore the mitigated selection pressure by NPIs allows the observed trend of the lower peak viral load of the BA.1 subvariant (which provides benefits at the host individual level such as evading host immunity) in Fig. 4b, c. However, the current framework of our multi-level model cannot explain the evolution of a "stealth" phenotype showing a decreased peak viral load with a high proportion of asymptomatic infection. A further detailed analysis is required once numerous datasets of different Omicron subvariants, such as BA.2, BA.4, BA.5, BQ.1, and XBB, become available.

Our study acknowledges certain limitations. Firstly, we assumed that the incubation period, $T^*$, and the proportion of symptomatic infected persons, $f$, are given as "environment" parameters, meaning we ignored the relation between virus evolution and clinical phenotypes. We expect that this is a reasonable assumption because recent SARS-CoV-2 human challenge clearly showed no quantitative correlation between patients' time-series pattern of viral load and symptoms[50]. Second, coevolutionary dynamics between hosts and viruses may result in long-term non-equilibrium evolutionary dynamics. A recent theoretical study revealed that viruses that often show antigenic escape from host immunity evolve through non-equilibrium dynamics[53], indicating that transmissibility fitness $R_{TP}$ is no longer the appropriate measure to understand patterns of virus evolution. Although the current framework does not consider such coevolutionary dynamics explicitly, we successfully captured transmission in viral load dynamics throughout the pandemic, which suggests that in our case transmissibility potential is a useful proxy. This may be because SARS-CoV-2 mutates slowly enough (on average, SARS-CoV-2 evolves 3-4 times slower than influenza virus[54]) to consider the long-term evolutionary equilibrium as in the classic context of endemic diseases. Third, our empirical data analysis and the multi-level simulation model do not explicitly account for the influence of population-level immune status on the evolution of SASR-CoV-2. Our additional analysis (Figs. S6 and S7), taking into consideration individuals with prior immunity, can alleviate concerns that individual host immune status significantly affects infection dynamics. However, the collective immune status at the population level can also shape the environment in which the virus evolves[55,56]. While strictly demonstrating the influence of population-level immune status on viral evolution through empirical data analysis may pose challenges, we maintain that our research offers significant insights into the potential interplay between human behavior changes and viral evolution.

The last but not least caveat for our modeling is the possibility that the evolutionary speed of SARS-CoV-2 and other potential selection pressures could generate the same evolutionary pattern presented here. The speed of evolution of a viral pathogen depends not only on the background mutation rate but also the virus generation time, the duration of infection, the number of variants that develop during the infection of an individual, etc[5]. If mutations occur too slowly, the virus prevalence decays prior to the appearance of a beneficial mutation that makes transmissibility higher. On the other hand, if mutations occur too rapidly, the pathogen evolution becomes volatile and, once again, the virus fails to spread[57]. Thus, mutation-selection balance is not negligible when discussing the evolutionary and epidemic outcomes in the future. We assumed that isolation truncated transmissibility time-series so that an earlier peak was

selected for in a plausible way. However, there may be another selection pressure to increase the number of secondary infections by increasing the initial slope of viral load dynamics, which also leads to an earlier peak. In general, which functional mutations can occur, how frequently they can occur, and how favorable they are compared with other strains should all be combined to predict the future trajectory of COVID-19.

In conclusion, using longitudinal viral load data, we demonstrated that an earlier and higher peak viral load but a shorter duration were selected for during SARS-CoV-2 evolution. We hypothesized that this evolution of SARS-CoV-2 infection dynamics might be associated with non-pharmacological interventions. However, it was challenging to strictly substantiate this hypothesis with current available data. Thus, we designed a probabilistic multi-level model, and our simulations using this model can inform hypotheses about the mechanisms that underlie virus evolution under isolation according to clinical phenotypes. Because NPIs are among the best ways of controlling a pandemic when a vaccine is not yet available, isolation as a strong NPI might be the first choice for a primary mitigation strategy for current and future emerging infectious disease. The impact of human behavior change on patterns of virus evolution must be considered when evaluating future scenarios of COVID-19. It would also be essential to investigate evolution considering economically preferred interventions, such as social distancing and mask-wearing, as well as isolation and combinations of these measures[58]. Although several types of COVID-19 vaccine are now available, many breakthrough infections are still being observed. Therefore, NPIs still play an important role in controlling the COVID-19 pandemic. It is also expected that new variants of concern after Omicron will become dominant. Thus, assuming that new variants and other pathogens will emerge in the future, it is important to consider the evolution of pandemic viruses when shaping public health strategies.

## Methods

### Study data

We searched for longitudinal viral load data from COVID-19 patients categorized with pre-Alpha, Alpha, Delta, and Omicron variants in PubMed and Google Scholar. Data meeting the following criteria were used to estimate parameters of the viral dynamics model: (1) viral loads were assessed at two time points or more and (2) samples were collected from the upper respiratory tract, such as from the nose or pharynx. Because all the data were extracted from published papers, ethics approval was not required in this study.

Four papers were identified that met the inclusion criteria. In total, viral load data from 86 patients infected with pre-Alpha variants (non-VOI/VOCs types), 59 Alpha variant patients, 80 Delta variant patients, and 49 Omicron (BA.1) variant patients were used for parameter estimation of the viral dynamics model. Three studies were from the USA and the other was from the UK (Table S1). First, viral load data from two previously published papers were used[14,31]. All patients' SARS-CoV-2 infections were by pre-Alpha variants. One case was reported from the USA and the other was from the UK. The original number of patients was 68 from the USA and 49 from the UK. Among those, data for 29 patients from the USA and 2 from the UK were excluded because of insufficient data points. For the second dataset consisting of the Alpha variant, two published papers were used[31,32]. For those studies, data for 22 patients from the USA and 37 from the UK were accepted; data from 14 patients in the USA and 2 in the UK were not included because of insufficient data points. Third, data for 64 patients from the USA[32,41] and 16 from the UK[31] were used for the Delta variant analysis. Originally, 26 additional data were in the USA dataset, but these were not used because there were two data points or fewer. Lastly, data from 49 patients collected in the USA[41] were used for the Omicron variant analysis.

## Modeling SARS-CoV-2 infection dynamics

To describe SARS-CoV-2 dynamics among susceptible target cells, we used a simple mathematical model for virus infection dynamics to define viral load for each patient:

$$\frac{dT(t)}{dt} = -\beta T(t)V(t) \tag{1}$$

$$\frac{dI(t)}{dt} = \beta T(t)V(t) - \delta I(t) \tag{2}$$

$$\frac{dV(t)}{dt} = pI(t) - cV(t) \tag{3}$$

The variables $T(t)$, $I(t)$, and $V(t)$ are the number of uninfected target cells, virus-producing infected cells, and the amount of virus at time $t$ since infection, respectively. The parameters $\beta$, $\delta$, $p$, and $c$ represent the rate constant for virus infection, the death rate of infected cells, the viral production rate, and the clearance rate of the virus, respectively.

To estimate the parameter values from the longitudinal viral load data of COVID-19 patients, we further simplified Eqs. (1−3). Since the clearance rate of the virus ($c = 20$ is fixed here) is typically much larger than the death rate of the infected cells in vivo, we made a quasi-steady state (QSS) assumption, $dV(t)/dt = 0$, and obtained $V(t) = pI(t)/c$, which yields $I(t) = cV(t)/p$. Then, substituting this equation into Eq. (2), we have

$$\frac{dV(t)}{dt} = \frac{p\beta}{c}T(t)V(t) - \delta V(t)$$

Furthermore, we replaced $T(t)$ by the proportion of target cells remaining at time $t$, that is, $f(t) = T(t)/T(0)$, where $T(0)$ is the initial number of uninfected target cells. Accordingly, we obtained the following simplified mathematical model[15–18], which we employed to analyze the viral load data in this study:

$$\frac{df(t)}{dt} = -\beta f(t)V(t) \tag{4}$$

$$\frac{dV(t)}{dt} = \gamma f(t)V(t) - \delta V(t) \tag{5}$$

where $\gamma = p\beta T(0)/c$ corresponds to the maximum viral replication rate under the assumption that target cells are continuously depleted during the course of infection. Thus, $f(t)$ is equal to or less than 1 and continuously declines.

## Nonlinear mixed effect model and test of the model fit

In our analyses, the variable $V(t)$ in Eq. (5) corresponds to the viral load for SARS-CoV-2. We separately analyzed two viral load datasets including the pre-Alpha, Alpha, and Delta variants and the Delta and Omicron variants because of different phases of virus evolution (see above). Note that the data for the Delta variant are common. We fixed values $V(t)$ and $T(t)$ at the time of infection as $10^{-2}$ and $1.33 \times 10^5$, respectively[33]. To fit the patient's viral load data, we used MONOLIX 2021R2 (www.lixoft.com), implement maximum likelihood estimation of parameters in nonlinear mixed effect model. The nonlinear mixed effect model allows a fixed effect as well as interpatient variability. This method estimates each parameter $\theta_i (= \theta \times e^{\eta_i})$ for each individual where $\theta$ is a fixed effect, and $\eta_i$ is a random effect, and which obeys a Gaussian distribution with mean 0 and standard deviation $\Omega$. Here we used lognormal distributions as prior distributions of parameters to guarantee the positiveness (i.e., negative values do not biologically make sense). In parameter estimation,

since time 0 on the original data set is the peak viral load time, we also estimated time from infection to peak viral load (corresponding to $T_p$ in Table S2 and S5) along with other parameters. We used the SARS-CoV-2 variants as a categorical covariate in estimating the parameters $\beta$, $\delta$, $\gamma$ (as well as $p = \gamma c / \beta T(0)$), and $T_p$, which provide the lowest AIC. Applying a stochastic approximation expectation-approximation algorithm and empirical Bayes method, the population parameters and individual parameters were estimated, respectively. The estimated fixed parameters and the initial values are listed in Table S2 for the pre-Alpha, Alpha, and Delta variants, and in Table S5 for the Delta and Omicron variants, respectively. The viral load curve using the best-fit parameter estimates for each individual is shown with the data in Figs. S1 and S8.

## Multi-level modeling for virus evolution

We explored how the time-series patterns of viral load, including the duration of viral shedding and the amount and timing of peak viral load of COVID-19 patients, which are all characterized by SARS-CoV-2 infection dynamics, change during the process of SARS-CoV-2 evolution. To this purpose, we here developed a probabilistic multi-level population dynamics model by coupling a population-level virus transmission model and the individual-level virus infection model (e.g., ref. 34).

First, as an individual-level virus infection model, we employed Eqs. (1−3) instead of Eqs. (4, 5). This is because all parameters in Eqs. (1−3) can be estimated from the data fitting of Eqs. (4, 5) to the long-itudinal viral load of SARS-CoV-2 for each variant. We interestingly found a positive correlation between the estimated death rate of infected cells, $\delta$, and the viral production rate $p$ (see Fig. S10). To mimic viral cytopathogenesis depending on the viral replication level, we defined $\delta = \delta_{\max}p/(p + p_{50})$, that is, an increasing Hill function of $p$. The parameters $\delta_{\max}$ and $p_{50}$ are the maximum value of $\delta$ and the viral production rate satisfying $\delta = \delta_{\max}/2$, respectively, and are estimated as summarized in Table S3. The dynamics of viral load calculated by this individual-level virus infection model will be translated into a probability of transmission and used in the following population-level model.

Next, as the population-level virus transmission model, we calculated the total number of secondary cases generated throughout the infectious period (corresponding to the duration of viral shedding, $D$) as $R_{TP}$, called "transmission potential". One simple assumption is that the viral population will eventually be dominated by the virus with the largest $R_{TP}$. On each day, the number of secondary cases is considered as the multiplication of the number of encounters (i.e., the contact number, $C(t)$) and the transmission probability, $\rho(t)$, and therefore $R_{TP}$ is defined as below (and explain its formulation below) (see Fig. 2):

$$R_{TP} = \sum_{t=0}^{D}(C(t)\rho(t)) \tag{6}$$

To mimic a daily contact history in Eq. (6), we assume the number of contacts on any day is drawn from a negative binomial distribution. Since a negative binomial distribution is identical to a Poisson distribution where the mean parameter $\lambda$ follows a gamma distribution, the daily contact numbers $C(t)$ of a focal infectious individual at time $t$ are generated by sampling from the following distributions[59]:

$$\lambda \sim gamma(k, \theta),$$

$$C(t) \sim P_o(\lambda),$$

where $k$ and $\theta$ are the shape parameter and the scale parameter, respectively (see Table S3). We note that $k$ influences the skewness, and $\theta$ influences the variance of the distribution, respectively. The

mean contact number through the history is the product of $k$ and $\theta$, that is, $k\theta$.

Once fixing the daily contact number, $R_{TP}$ (i.e., Eq. (6)) is dependent on $\rho(t)$ and $D$, meaning the parameters appearing in Eqs. (1–3). To evaluate the daily transmissibility of an infected individual through the duration of viral shedding, we assumed that a probability of transmission (per contact with a susceptible individual), $\rho(t)$, depends on viral load. As reported in[60], we simply employed the step function as the range of viral load; thus, the transmission probability $\rho(t)$ at time $t$ is described by

$$\rho(t) = \begin{cases} 0.00 & \text{if } \log_{10} V(t) < 5, \\ 0.12 & \text{if } 5 \le \log_{10} V(t) < 7, \\ 0.15 & \text{if } 7 \le \log_{10} V(t) < 10, \\ 0.24 & \text{if } 10 \le \log_{10} V(t). \end{cases}$$

For example, $\rho(t)$ for SARS-CoV-2 variants are calculated based on Eq. (3) in Fig. 1f.

In addition, since there is a positive correlation between the duration of viral shedding ($D$) and the cumulative log-transformed viral load ($V_{\text{total}}$) for COVID-19 patients observed in Fig. 1e, we defined

$$D = \frac{D_{\max} V_{\text{total}}^{D_K}}{V_{50}^{D_K} + V_{\text{total}}^{D_K}} \tag{7}$$

$$V_{\text{total}} = \int_0^{T_{\text{DL}}} \log_{10} V(t)\, dt \tag{8}$$

where $T_{\text{DL}}$ is the time for the viral load to reach the detection limit (DL=1 is fixed as an arbitrary value). The parameters $D_{\max}$, $V_{50}$ and $D_k$ are the maximum duration of viral shedding, the total viral load at which the duration is half of its maximum, and the steepness at which duration increases with increasing viral load, respectively, and are estimated as summarized in Table S3.

We hereafter focus on the dependence of the rate constant for virus infection $\beta$ and the viral production rate $p$ in Eqs. (1–3), since the death rate of infected cells $\delta$ is function of $p$ and the other parameters or initial values are fixed or common as explained above. We calculate $R_{TP}$ in various patterns of ($\beta$,$p$) by Eqs. (1–3, 6), and thus the transmissibility fitness landscape of $R_{TP}$ is constructed as a function of $\beta$ and $p$.

### Evolution of virus to increase transmission potential

When calculating the transmission potential $R_{TP}$, the focal infectious individuals are assigned one of two properties: symptomatic or asymptomatic, which is randomly assigned by a constant probability $f$ or $1 - f$. We assume $f$ to be independent of the viral load dynamics, because there is no significant difference in viral load between asymptomatic and symptomatic infection as discussed in[61,62]. We here simply assumed the transmission chains from the focal infectious individuals are perfectly inhibited by isolation after symptom onset. That is, if the focal infectious individual is symptomatic, we stopped calculating the cumulative number of secondary cases at symptom onset. Symptom onset starts after $T^*$ days of infection, which represents the incubation period. Otherwise, $R_{TP}$ is calculated using the whole duration of viral shedding of the focal asymptomatic infectious individual.

We analyze the virus evolutionary dynamics on the transmissibility potential landscape. The set of the value of ($\beta$,$p$) are explored in the sense of maximizing $R_{TP}$, using a genetic algorithm, GA[63] (see Algorithm S1 and Table S4 for parameters used in GA). Initially, GA starts with an arbitrary set of parameters ($\beta$,$p$) and initial values (Initialize), and calculates $R_{TP}$ as transmissibility fitness (Evaluate). Over successive iterations, the values of ($\beta$,$p$) with higher $R_{TP}$ are

selected to produce the next solutions (selectSolutions). Also, the two highest-performing solutions are always chosen for the next iteration (Elitism). To find better values of ($\beta$,$p$), solutions are recombined (crossover), and each parameter is slightly altered by adding a random variable drown from a Uniform distribution with a pre-determined range (Mutate), which corresponds to the rate of evolutionary change of the virus. Finally, by varying $T^*$ and $f$, we explore a better set of values ($\beta$,$p$) that maximizes $R_{TP}$. Note that the set of values at the end of the GA run is not necessarily a global optimum. These iterations can be considered as the virus "evolving" its properties of $\beta$ and $p$ toward an optimal solution so that the virus can increase its own fitness in the evolutionary history.

### Reporting summary

Further information on research design is available in the Nature Portfolio Reporting Summary linked to this article.

## Data availability

All viral load data that support the findings of this study are available at Zenodo[64] (https://zenodo.org/records/10031081). Source data are provided with this paper.

## Code availability

All analyses were performed with the software Python (version 3.10) and R (version 4.3.0). The analysis using nonlinear mixed effects model (including the algorithms for parameters estimation, such as Stochastic Approximation Expectation Maximization and Markov Chain Monte Carlo) was performed on MONOLIX 2021R2 (www.lixoft.com). Our code is publicly available at Zenodo[64] (https://zenodo.org/records/10031081).

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

## Acknowledgements
This study was supported in part by the National Research Foundation of Korea (NRF) grant funded by the Korea government (MSIT) (2022R1C1C2003637) (to K.S.K.); Grant-in-Aid for Transformative Research Areas A 22H05215 (to S.I.), JSPS Scientific Research (KAKENHI) B 18H01139 (to S.I.), 16H04845 (to S.I.), 21K15160 (to R.Y.), Scientific Research in Innovative Areas 20H05042 (to S.I.); JSPS Overseas Research Fellowships (to R.Y.); ACT-X JPMJAX22AK (to R.Y.); AMED CREST 19gm1310002 (to S.I.); AMED Development of Vaccines for the Novel Coronavirus Disease, 21nf0101638s0201 (to S.I.); AMED Japan Program for Infectious Diseases Research and Infrastructure, 20wm0325007h0001 (to S.I.), 20wm0325004s0201 (to S.I.), 20wm0325012s0301 (to S.I.), 20wm0325015s0301 (to S.I.); AMED Research Program on HIV/AIDS 22fk0410052s0401 (to S.I.); AMED Research Program on Emerging and Re-emerging Infectious Diseases 20fk0108140s0801 (to S.I.), 21fk0108428s0301 (to S.I.); AMED Program for Basic and Clinical Research on Hepatitis 21fk0210094 (to S.I.); AMED Program on the Innovative Development and the Application of New Drugs for Hepatitis B 22fk0310504h0501 (to S.I.); AMED Strategic Research Program for Brain Sciences 22wm0425011s0302; AMED JP22dm0307009 (to K.A.); JST MIRAI JPMJMI22G1 (to S.I.); Moonshot R&D JPMJMS2021 (to K.A. and S.I.) and JPMJMS2025 (to S.I.); Institute of AI and Beyond at the University of Tokyo (to K.A.); Shin-Nihon of Advanced Medical Research (to S.I.); SECOM Science and Technology Foundation (to S.I.). We would like to thank Sarah Otto for her helpful feedback on earlier versions of the manuscript.

## Author contributions
S.I. and R.Y. designed the research. J.S., H.P., K.S.K., R.K., S.C., L.R.T., J.W., Y.D.J., W.S.H., R.N.T., K.A., S.I. and R.Y. carried out the computational analysis. S.I. and R.Y. supervised the project. All authors contributed to writing the manuscript.

## Competing interests
Authors declare no competing interests.
