## [Peer Review File · Nature Communications]

Reviewers' Comments:

Reviewer #3:

Remarks to the Author:

The authors have made a great effort to improve the manuscript according to the comments made by all three reviewers. As in any theoretical study that incorporates real data, even this study has its limitations that arise from the type of data that are being analyzed. However, these limitations have been addressed by the authors to my satisfaction.

At this point, I have no further questions or suggestions and am in favor of publishing the study in Nature Comms or any other journal.

Reviewer #4:

Remarks to the Author:

The authors present a compelling hypothesis that isolation of symptomatic individuals creates a selective environment which influences the evolution of viral kinetics leading to increased peak viral load and decreased time to peak load as well as decreased duration of infectivity. I believe this work to be of broad interest in two ways. First, the presentation of the hypothesis itself (even if straightforward in retrospect) represents a creative and uncommon view on a topic of enormous interest. Additionally, the aggregate kinetic data presented is in relatively short supply and I hope will be utilized by other groups for additional analysis.

I understand the work has been under review for some time and I believe the authors made an excellent effort to address the reviewer comments made from the previous round. I have no remaining concerns that were raised but not addressed in the previous round of review with one exception (see below).

I do wish to emphasize two points. First, globally, nonpharmaceutical interventions including isolation were implemented during the acute phase of the SARS-CoV-2 pandemic, during which the default expectation is that the rate of adaptive evolution would be high (at least at its peak relative to the endemic phase). While the authors provide in my view close to the best available controls to identify the impact of public health policy independent of other underlying environmental shifts, I am not convinced the data availability is great enough to determine causality. I want to emphasize that I find the model presented and the demonstration that isolation could, in principle, exert great enough selective pressures to impact the evolutionary landscape compelling. I simply do not believe the data available presents the opportunity to falsify the hypothesis that in the absence of nonpharmaceutical intervention, the same events in the molecular evolution of the virus would have been observed.

This is additionally complicated by the fact that a variable immune landscape over the course of the pandemic (through the accumulation of prior infections and vaccinations) almost certainly impacted the selective environment (although even this may be impossible to strictly demonstrate). This was addressed at least once in the first round of review, "In particular, I was confused by the lack of discussion in either the Results or Methods section on the effect of prior immunity (either caused by vaccination or infection)." In response, the authors conducted additional modelling and data analysis. I have no critical comments regarding the modelling. With respect to the additional empirical results presented, the authors write, "we further analyzed SARS-CoV-2 infection dynamics, including data of patients who were excluded from previous analysis for reasons of prior immunity. 14 Alpha variant patients and 54 Delta variant patients with prior immunity were added in analysis. In this analysis, we obtained conclusions consistent with previous results: the duration of viral shedding decreased as SARS-CoV2 evolved."

This alleviates the concern (which may be what the original reviewer intended to express) that the immune status of an individual host impacts infection kinetics and if the same viral genotypes observed later in the pandemic were to cause infections in naïve populations, the differences in kinetics reported may be reduced. This does not address what I view to be the larger issue here, that the average immune status of the population modifies the selective pressures which almost surely shapes viral kinetics. [I may be biased as I have written on this topic: <https://www.ncbi.nlm.nih.gov/pmc/articles/PMC8406440/>, <https://www.ncbi.nlm.nih.gov/pmc/articles/PMC9040817/>] The authors acknowledge this limitation in the response to this comment, "Nevertheless, we maintain that our research provides significant insights into the potential relationship between human behavior change and viral evolution, although it is impossible to prove our hypothesis in real world," and I agree with their self-assessment of significance.

My second point concerns the presentation and potential future development of the model. As I state above, I believe the model developed is insightful and technically correct. I also believe, however, the presentation could be modified to explicitly discuss the impact of including or excluding trade-off hypotheses describing constraints on viral life history characteristics. The authors write, "From the perspective of virus evolution, which maximizes the basic/effective reproduction number...the evolution of a higher proportion of asymptomatic infection is always favored because transmissions are not limited when those infected individuals are not isolated. This means that once the higher proportion of asymptomatic infection evolves, other viral phenotypic changes designed to avoid isolation will be suppressed. In other words, viral phenotypes showing slower and/or lower peak viral load can evolve (see Extended Data Fig. 3)." This perspective assumes the evolution of virulence and that of infectivity may often be decoupled in realistic scenarios. [I am in favor of this perspective but again may be biased based on my own prior work on this topic: <https://www.ncbi.nlm.nih.gov/pmc/articles/PMC8381341/>] This is not, I believe, the consensus opinion, however, which instead assumes both infectivity and virulence increase with viral load <https://pubmed.ncbi.nlm.nih.gov/30734920/>. Of course immune-modulating adaptations, perhaps including those observed in the Omicron variant relative to prior variants <https://pubmed.ncbi.nlm.nih.gov/36175424/>, complicate this picture.

The success of an evolutionary strategy to increase viral titers early in the course of infection to thwart isolation depends on the ability to mitigate the hastening or severity of symptoms. If ramping up viral load faster also hasten symptom onset, this may represent a neutral or even deleterious phenotype as it shortens the duration of infectivity prior to isolation. With this in mind, I would be very interested in seeing the authors further develop the model to assess the predicted evolutionary impact of other nonpharmaceutical interventions principally mask wearing and the dissemination of test positivity information which impacts host contact rates (although I expect this to be beyond the scope of the present manuscript). Recently, testing with self isolation was demonstrated to be an economically preferred intervention for a generalized model system <https://pubmed.ncbi.nlm.nih.gov/37210388/>.

Overall I find this work to be compelling, creative, and of very broad interest.

Sincerely,

Nash Rochman (review requested 08/31/23; response returned 09/19/23 – my apologies for the delay)

Response to reviewer comments

Junya Sunagawa, Hyeongki Park, Kwang Su Kim, Ryo Komorizono, Sooyoun Choi, Lucia Ramirez Torres, Joohyeon Woo, Yong Dam Jeong, William S Hart, Robin N. Thompson, Kazuyuki Aihara, Shingo Iwami and Ryo Yamaguchi

October 24, 2023

We would like to thank the reviewers for their thorough reading of the manuscript and their thoughtful comments and suggestions. In this letter, we present the reviewers' comments in black and provide our responses in red. All the changes made in response to the reviewers' comments in NCOMMS-23-38617-T have been incorporated in the revised version of the manuscript. We believe that the manuscript has been considerably improved by these changes. All coauthors agreed with these changes.

Reviewer #3

The authors have made a great effort to improve the manuscript according to the comments made by all three reviewers. As in any theoretical study that incorporates real data, even this study has its limitations that arise from the type of data that are being analyzed. However, these limitations have been addressed by the authors to my satisfaction. At this point, I have no further questions or suggestions and am in favor of publishing the study in Nature Comms or any other journal.

Reply to Comments:

We are deeply appreciative of your positive evaluation of our paper.

Reviewer #4

The authors present a compelling hypothesis that isolation of symptomatic individuals creates a selective environment which influences the evolution of viral kinetics leading to increased peak viral load and decreased time to peak load as well as decreased duration of infectivity. I believe this work to be of broad interest in two ways. First, the presentation of the hypothesis itself (even if straightforward in retrospect) represents a creative and uncommon view on a topic of enormous interest. Additionally, the aggregate kinetic data presented is in relatively short supply and I hope will be utilized by other groups for additional analysis. I understand the work has been under review for some time and I believe the authors made an excellent effort to address the reviewer comments made from the previous round. I have no remaining concerns that were raised but not addressed in the previous round of review with one exception (see below).

Reply to Comments:

We appreciate your evaluation of the significance of our current study and the revision process. We have addressed the concerns based on your constructive comments, which have further contributed to improving the manuscript.

I do wish to emphasize two points. First, globally, nonpharmaceutical interventions including isolation were implemented during the acute phase of the SARS-CoV-2 pandemic, during which the default expectation is that the rate of adaptive evolution would be high (at least at its peak relative to the endemic phase). While the authors provide in my view close to the best available controls to identify the impact of public health policy independent of other underlying environmental shifts, I am not convinced

the data availability is great enough to determine causality. I want to emphasize that I find the model presented and the demonstration that isolation could, in principle, exert great enough selective pressures to impact the evolutionary landscape compelling. I simply do not believe the data available presents the opportunity to falsify the hypothesis that in the absence of nonpharmaceutical intervention, the same events in the molecular evolution of the virus would have been observed.

Reply to Comments:

We appreciate for your comment regarding data availability. We acknowledge that data availability in this study may not be sufficient to strictly demonstrate our assumption. To discuss about this concern, we have added a sentence in the Discussion section (Page 20, Lines 418-423).

This is additionally complicated by the fact that a variable immune landscape over the course of the pandemic (through the accumulation of prior infections and vaccinations) almost certainly impacted the selective environment (although even this may be impossible to strictly demonstrate). This was addressed at least once in the first round of review, “In particular, I was confused by the lack of discussion in either the Results or Methods section on the effect of prior immunity (either caused by vaccination or infection).” In response, the authors conducted additional modelling and data analysis. I have no critical comments regarding the modelling. With respect to the additional empirical results presented, the authors write, “we further analyzed SARS-CoV-2 infection dynamics, including data of patients who were excluded from previous analysis for reasons of prior immunity. 14 Alpha variant patients and 54 Delta variant patients with prior immunity were added in analysis. In this analysis, we obtained conclusions consistent with previous results: the duration of viral shedding decreased as SARS-CoV2 evolved.”

This alleviates the concern (which may be what the original reviewer intended to express) that the immune status of an individual host impacts infection kinetics and if the same viral genotypes observed later in the pandemic were to cause infections in naïve populations, the differences in kinetics reported may be reduced. This does not address what I view to be the larger issue here, that the average immune status of the population modifies the selective pressures which almost surely shapes viral kinetics. [I may be biased as I have written on this topic: <https://www.ncbi.nlm.nih.gov/pmc/articles/PMC8406440/>, <https://www.ncbi.nlm.nih.gov/pmc/articles/PMC9040817/>] The authors acknowledge this limitation in the response to this comment, “Nevertheless, we maintain that our research provides significant insights into the potential relationship between human behavior change and

viral evolution, although it is impossible to prove our hypothesis in real world,” and I agree with their self-assessment of significance.

Reply to Comments:

We greatly appreciate the recognition of the importance of our results. As pointed out by the reviewer, although our current methods do not reflect the average immune status of the population, we still believe our results provide valuable insights. To clarify this point, we have discussed it as an additional limitation in the Discussion section of the main text, along with suggested references (Page 19, Lines 393-401).

My second point concerns the presentation and potential future development of the model. As I state above, I believe the model developed is insightful and technically correct. I also believe, however, the presentation could be modified to explicitly discuss the impact of including or excluding trade-off hypotheses describing constraints on viral life history characteristics. The authors write, “From the perspective of virus evolution, which maximizes the basic/effective reproduction number...the evolution of a higher proportion of asymptomatic infection is always favored because transmissions are not limited when those infected individuals are not isolated. This means that once the higher proportion of asymptomatic infection evolves, other viral phenotypic changes designed to avoid isolation will be suppressed. In other words, viral phenotypes showing slower and/or lower peak viral load can evolve (see Extended Data Fig. 3).” This perspective assumes the evolution of virulence and that of infectivity may often be decoupled in realistic scenarios. [I am in favor of this perspective but again may be biased based on my own prior work on this topic: <https://www.ncbi.nlm.nih.gov/pmc/articles/PMC8381341/>] This is not, I believe, the consensus opinion, however, which instead assumes both infectivity and virulence increase with viral load <https://pubmed.ncbi.nlm.nih.gov/30734920/>. Of course immune-modulating adaptations, perhaps including those observed in the Omicron variant relative to prior variants <https://pubmed.ncbi.nlm.nih.gov/36175424/>, complicate this picture.

Reply to Comments:

As reviewer pointed out, we assumed that the evolution of virulence and that of infectivity may be decoupled. However, we also totally agree with the reviewer’s perspective, that this assumption regarding interplay between these two factors is not universally agreed upon. We greatly appreciate the insightful feedback provided by the reviewer, and recognize the opportunity to further elaborate on the interrelationship between evolution of virulence and that of infectivity in our work. To convey this significance to the readers, we have discussed this

perspective incorporating the suggested references in the Discussion section of the main text during our revision (Page 18, Lines 358-361).

The success of an evolutionary strategy to increase viral titers early in the course of infection to thwart isolation depends on the ability to mitigate the hastening or severity of symptoms. If ramping up viral load faster also hasten symptom onset, this may represent a neutral or even deleterious phenotype as it shortens the duration of infectivity prior to isolation. With this in mind, I would be very interested in seeing the authors further develop the model to assess the predicted evolutionary impact of other nonpharmaceutical interventions principally mask wearing and the dissemination of test positivity information which impacts host contact rates (although I expect this to be beyond the scope of the present manuscript). Recently, testing with self isolation was demonstrated to be an economically preferred intervention for a generalized model system <https://pubmed.ncbi.nlm.nih.gov/37210388/>. Overall I find this work to be compelling, creative, and of very broad interest.

Reply to Comments:

We greatly appreciate your suggestion for the intriguing potential extensions regarding our model. Indeed, if the viral load peak increases, there may be different drawbacks for the virus in return for evading isolation among infected individuals. Our current study assumed strong Non-Pharmaceutical Interventions (NPIs) such as isolation. However, in future research, we also acknowledge the possibility of exploring phenotypic evolution induced by moderate NPIs, like social distancing and mask-wearing. We have briefly touched upon this direction in the Discussion section (Page 21, Lines 427-429). Thank you once again for your invaluable insights.